# The Aquaporin 3 Promoter Polymorphism −1431 A/G is Associated with Acute Graft Rejection and Cytomegalovirus Infection in Kidney Recipients Due to Altered Immune Cell Migration

**DOI:** 10.3390/cells9061421

**Published:** 2020-06-08

**Authors:** Katharina Rump, Tim Rahmel, Anna-Maria Rustige, Matthias Unterberg, Hartmuth Nowak, Björn Koos, Peter Schenker, Richard Viebahn, Michael Adamzik, Lars Bergmann

**Affiliations:** 1Klinik für Anästhesiologie, Intensivmedizin und Schmerztherapie, Universitätsklinikum der Ruhr Universität Bochum Knappschaftskrankenhaus Bochum, 44801 Bochum, Germany; Tim.Rahmel@ruhr-uni-bochum.de (T.R.); AnnaRustige@web.de (A.-M.R.); Matthias.Unterberg@kk-bochum.de (M.U.); hartmuth.nowak@kk-bochum.de (H.N.); bjoern.koos@rub.de (B.K.); michael.adamzik@kk-bochum.de (M.A.); lars.bergmann@kk-bochum.de (L.B.); 2Chirurgische Universitätsklinik, Universitätsklinikum Knappschaftskrankenhaus Bochum, 44892 Bochum, Germany; peter.schenker@rub.de (P.S.); richard.viebahn@kk-bochum.de (R.V.)

**Keywords:** AQP3, aquaporin 3, polymorphism, kidney transplantation, AQP3 −1431 A/G, rs3860987, CMV infection, cytomegalovirus, rejection, migration

## Abstract

Major complications after kidney transplantation are graft rejection and cytomegalovirus (CMV) infection, which are related to T-cell function, which depends on aquaporin 3 (AQP3) expression. The impact of the AQP3 A(−1431)G promoter polymorphism in kidney transplant recipients was unelucidated and we explored the effect of AQP3 polymorphism on immune cell function and its association with graft rejection and CMV infection in 237 adult patients within 12 months after transplantation. AQP3 promoter polymorphism was molecular and functional characterized. Kaplan–Meier plots evaluated the relationship between genotypes and the incidence of CMV infection and graft rejection. AQP3 A(−1431)G A-allele was associated with enhanced immune cell migration and AQP3 expression in T-cells. The incidences of rejection were 45.4% for the A-allele and 27.1% for G-allele carriers (*p* = 0.005) and the A-allele was a strong risk factor (hazard ratio (HR): 1.95; 95% CI: 1.216 to 3.127; *p* = 0.006). The incidences for CMV infection were 21% for A-allele and 35% for G-allele carriers (*p* = 0.013) and G-allele was an independent risk factor (*p* = 0.023), with a doubled risk for CMV infection (HR: 1.9; 95% CI: 1.154 to 3.128; *p* = 0.012). Hence, A-allele confers more resistance against CMV infection, but susceptibility to graft rejection mediated by T-cells. Thus, AQP3-genotype adapted management of immunosuppression and antiviral prophylaxis after kidney transplantation seems prudent.

## 1. Introduction

Acute kidney injury is a common complication, which occurs as a response to severe, acute, or chronic diseases, such as sepsis, diabetes, or high blood pressure [1]. Despite treatment of the underlying cause, in some cases, kidney function cannot be recovered and an end-stage renal disease arises [2]. Kidney transplantation is currently the best therapy option for patients with end-stage renal disease [3]. The success of kidney transplantation depends on many factors, such as age, ethnicity, human leukocyte (HLA)-mismatch, and ischemia time [4]. Despite growing experience with kidney transplantation throughout the past decades, the incidence of severe complications remains high, which reflects high interindividual variability in the postoperative course of disease [5]. Two major complications are acute graft rejection and cytomegalovirus (CMV) infection, which greatly affect graft function [6] and might also influence mortality rates [7,8]. Both complications are related to T-cell function [9,10]. High T-cell activity after transplantation can cause increased rejection, whereas diminished T-cell function can increase the risk of CMV infection [11,12]. Despite immunomodulatory therapy, it is difficult to maintain an immune reaction balance between effective avoidance of CMV infection and immune tolerance to the graft [12]. In fact, observed complications and interindividual differences cannot be fully explained by classical risk factors alone and seem inter alia to depend on genetic variations [6,13,14]. Regarding genetic variations, several candidate genes such as those for cytochrome P450 and interleukins have been implicated [15,16]. Aquaporins (AQPs) are related to immune cell and kidney function [17] and thus, might be of special interest. In the last years, the effects of AQPs on both graft rejection and CMV infection have garnered attention in immunological responses studies [18]. It was demonstrated that AQPs play an important role in key mechanisms of inflammation including immune cell migration and proliferation [19,20]. Hence, high AQP expression could cause increased rejection due to high immunological activity and low AQP expression can lead to CMV infection due to insufficient immunological activity [18]. The AQP family consists of 13 identified proteins [21]. In the kidney, aquaporin 3 (AQP3) is particularly important, as it is localized in the epithelial cells of the collecting duct and connecting tubule [22]. Moreover, AQP3 is required for T cell migration [23]. Given that increased AQP3 expression linked to elevated T-cell migration, interindividual differences in T-cell function might depend on AQP3 expression [23]. Differential AQP3 expression could result from genetic variants in the AQP3 promoter and might affect kidney function or immunological reactions via altering T-cell function after transplantation. Interestingly, AQP3 contains the single nucleotide promoter polymorphism (SNP, rs3860987), AQP3 A(−1431)G, which is described in Single Nucleotide Polymorphism Database (dbSNP) (www.ncbi.nlm.nih.gov/projects/SNP) and has only been studied in one clinical trial [24]. This SNP might affect AQP3 expression, which in turn could influence outcome after kidney transplantation. Therefore, we tested the hypothesis that the AQP3 A(−1431)G promoter polymorphism is functionally active, alters immune cell migration and AQP3 expression in T-cells, and is associated with acute rejection and CMV infection after kidney transplantation.

## 2. Materials and Methods

### 2.1. Study Design

The aim of the present study was molecular and functional characterization of the AQP3 polymorphism AQP3 A(−1431)G (rs3860987). For this purpose, a combined in vivo and in vitro study was conducted. First, in vitro molecular characterization of the AQP3 A(−1431)G polymorphism was carried out using luciferase reporter assay, electrophoretic mobility shift assay, and expression analysis of samples from healthy volunteers. For functional characterization, migration assay with samples from healthy volunteers was carried out. Finally, the results from the in vitro study were confirmed in vivo in a cohort of well-characterized kidney transplant recipients to examine the clinical impact of the polymorphism on pathophysiology after kidney transplantation, with special focus on biopsy-proven rejection and risk for CMV infection.

### 2.2. Cloning of the AQP3 Promoter Constructs

For functional characterization, the *AQP3* promoter was cloned into a reporter vector as follows. Cloning of *AQP3* promoter constructs was performed using the following primers (Eurofins Genomics, Ebersberg, Germany): AQP3_Prom_SE: TATAGGAGCGCTGGAGACAC and AQP3_Prom_AS: TCAGCCTAAGGGCATGTTGT. PCR products for different genotypes were ligated into the pGEMt-easy vector (Promega, Fitchburg, MA, USA). Subsequent deletion constructs produced by targeted digestion of products, using *EcoRI*, *EcoRV*, *AlwNI*, and *XhoI* (New England Biolabs, Ipswich, MA, USA), were ligated into the pGL4.10 reporter gene vector (Promega, Madison, WI, USA). Existence of the following promoter regions was confirmed by sequencing (Eurofins, Genomics): (nucleotide (nt)-873/nt-91, nt-367/nt-91, nt-474/nt-91, nt-1491/nt-1334 (−1431 A), and nt-1491/nt-1334 (−1431 G).

### 2.3. Luciferase Assay

Caco-2 and HeLa (origin Deutsche Sammlung von Mikroorganismen und Zellkulturen GmbH (DZMZ), Braunschweig, Germany) were used for luciferase assays. Cells were seeded in 96-well culture plates (15,000 HeLa/Caco-2 cells in 100 µL DMEM + 10% FKS). The following day, the cells were transfected using Lipofectamine 3000 (Invitrogen, Darmstadt, Germany) with 150 ng promoter construct in pGL4.10 or 50 ng pGL4.74 control vector. For specific transcription factor studies in silico analysis was performed using Genomatix software suit (www.genomatix.com) and patch (patch public 1.0 Pattern Search for Transcription Factor Binding Site, www.gene-regulation.com) for the analysis of putative transcription factor binding sites. This analysis revealed cAMP response element-binding protein (CREB) and specificity protein 1 (SP1) to putative bind to the AQP3 promoter. Hence, cells were transfected with 75 ng promoter construct in pGL4.10 or 25 ng pGL4.74 control vector and 100 ng of pcDNA3.1 overexpression vector (cAMP response element-binding protein (CREB)_pcDNA3.1, specificity protein 1 (SP1) pcDNA3.1, or empty pcDNA3.1). Cell culture medium was replaced 24 h after transfection with 75 µL fresh medium. Luciferase activity was measured with Dual-Glo Luciferase Assay System (Promega, Madison, WI, USA), following the manufacturer’s instructions.

### 2.4. Electrophoretic Mobility Shift Assay (EMSA) of Transcription Factor Binding

EMSA was carried out using EMSA buffer kit (Li-Cor, Bad Homburg, Germany). Nuclear extracts from Caco-2 were used (Nuclear Extraction Kit, Abcam, Cambridge, UK). Sequences of EMSA oligonucleotides (Eurofins Genomics, Ebersberg, Germany) were as follows: for AQP3 −1431 analysis: SE: −1431_SE: 5′-Agttcgaggctacagt(G/A)agctgtgattgca-3′ and −1431_As: 5′-tgcaatcacagct(C/T)actgtagcctcgaact-3′. Sense and antisense oligonucleotides were labeled with IR-dye 682 and hybridized by slow cooling after boiling to 100 °C. The probes were incubated with 10 μg nuclear extracts for 20 min at room temperature. Moreover, 100× excess unlabeled double-stranded oligonucleotide was added for the competition analysis. The samples were loaded on non-denaturing 6% polyacrylamide gel and electrophoresis was carried out. Bands were visualized in gel using Li-Cor Odyssey system according to the manufacturer’s instructions (LiCor). The experiments were performed four times in total.

### 2.5. Patient Samples and DNA Isolation

The study was reviewed and approved by the Ethics’ committee of the Ruhr Universität Bochum (reg. no. 4870-13). Between 2007 and 2014, patients, who received either kidney or combined pancreas-kidney transplantation at the Department of General Surgery of the University Hospital Knappschaftskrankenhaus Bochum (Bochum, Germany), were enrolled. Patients were recruited to donate a buccal swab for DNA extraction and the evaluation of *AQP3* SNP, A(−1431)G (rs3860987), after transplantation. For study inclusion, written informed consent was obtained from all 237 participating patients, according to the Declaration of Helsinki, good clinical practice guidelines, and applicable to local regulatory requirements. Demographic and clinical parameters such as age, bodyweight, height, underlying disease and days until acute rejection of the transplant were recorded upon study inclusion and patients were followed up for at least 1 year after organ transplantation (Table 1). All patients received immunosuppressive induction and maintenance therapy according to locally specific standard operating procedures, which included steroids combined with Anti-Thymocyte-Globulin (ATG) or interleukin-2 receptor antibodies (IL-2 RA) (Table 1), as well as risk-adapted perioperative and postoperative antiviral chemoprophylaxis with ganciclovir or valganciclovir. Patients were assigned to different risk groups according to their own pre-operative CMV status (Recipient (R), R^+^ = CMV positive, R^−^ CMV negative) and the donors (D) CMV-status (D^+^ = CMV positive, D^−^ CMV negative). In this context, 54 high-risk patients (D^+^/R^−^) received chemoprophylaxis for 6 months (except for one patient in this group with unknown or shorter duration), 132 medium-risk patients (D^+^/R^+^ and D^−^/R^+^) received prophylaxis for 3 months (except for 8 patients in this group with unknown or shorter duration), and 36 low-risk patients (D^−^/R^−^) received perioperative prophylaxis, for whom chemoprophylaxis was expanded to 3 months in 20 cases, for example, due to CMV-positive blood transfusions.

Delayed graft function was defined as the necessity for hemodialysis in the first week after surgery. When rejection was suspected, a biopsy of the kidney graft was performed primarily and graded according to the Banff classification [25], hence, rejection was defined by biopsy-proven acute rejection [26]. Routine surveillance for viral reactivation or infection comprised weekly determinations of CMV viremia based on whole blood samples, until hospital discharge from index-admission; and thereafter, continuing monthly or when clinically indicated. Additionally, all patients were screened for CMV infection at the 1-year follow up examination after kidney transplantation. CMV infection was defined as the detection of viral nucleic acid in accordance to the definition of Ljungman and colleagues [27]. In detail, surveillance for CMV infection was performed weekly during the first month following transplantation and monthly thereafter until the end of the first year or in cases of clinical suspicion. CMV DNA was evaluated using a commercially available PCR assay (Roche Ampliprep Assay; Cobas 4800; Roche Molecular Diagnostics, Pleasanton, CA, USA) as per the manufacturer’s instructions and calibrated to the World Health Organization International Standard for Human CMV. Any level of detection was considered positive. CMV disease and related entities (e.g., CMV pneumonia and CMV syndrome) were defined as the presence of CMV in the blood based on a local assay plus the presence of compatible symptoms, as described by Ljungman and colleagues [27], which include newly occurred malaise, fever > 38 °C, leukopenia and thrombocytopenia, elevation of hepatic transaminases to more than twice the standard values, or graft dysfunction. Treatment of CMV infection was performed using intravenous ganciclovir adjusted to the calculated glomerular-filtration rate and valganciclovir treatment.

DNA samples were isolated from buccal swabs using QiaAMP DNA Mini Kit (Qiagen, Hilden Germany). For genotyping rs3860987, named AQP3 A(−1431)G primers (MWG Eurofins, Ebersberg, Germany) were utilized for standard Polymerase chain reaction (PCR) using OneTaq^®^ Quick-Load^®^ 2X Master Mix with Standard Buffer (New England Biolabs, Frankfurt, Germany) with an annealing temperature of 60 °C and 30 min elongation time. The primers AQP3_1431_SE: CATCCATTCTGGAGGCTGAG and AQP3_1431_AS: CCCTATCTGAGGGGGATCAT resulted in a 158-bp PCR product. Endonuclease restriction was performed using *TspRI* (New England Biolabs, Frankfurt, Germany) resulting in two fragments (98 bp + 60 bp) for G-allele or one fragment (158 bp) for A-allele.

### 2.6. Measurement of AQP3 Expression in Immune Cells

After approval of the ethics committee of the Ruhr-Universität Bochum (ref: 17-5964-BR) and written informed consent, eight healthy volunteers were genotyped and 50 mL EDTA blood was collected. Blood was separated into peripheral blood mononuclear cells (PBMCs) using Ficoll density gradient centrifugation, neutrophils using MACS express Whole Blood Neutrophil Isolation Kit, or T-cells using MACS express Whole Blood Pan T cell Isolation Kit (Miltenyi Biotech, Bergisch Gladbach, Germany). PBMCs were stimulated with 200 mM 8-Br cAMP (Sigma-Aldrich, St Louis, MO, USA) or left unstimulated. Total RNA was extracted using the RNeasy kit (Qiagen, Hilden, Germany) and first strand cDNA synthesized from 1 µg RNA using QiaAMP reverse transcription kit (Qiagen). The following primers were used to perform real-time PCR: AQP3_RT_SE: 5′- GATGCAATCTGGCACTTCGC -3′ and AQP3_RT_AS: 5′- ACACACACGATAAGGGAGGC -3′, resulting in a 152-bp fragment. Primers used for the housekeeping gene, β-Actin, were as previously described [28]. Real-time PCR reaction was performed using GoTaq^®^ qPCR Master Mix (Promega, Madison, WI, USA). In addition, PCMCs, neutrophils, and T-cells were sonicated on ice in 200 µL PBS containing HALT cocktail. After quantification of total protein amount, AQP3 protein amount was determined using Human Aquaporin 3, Gill Blood Group (AQP3) ELISA Kit (Biomatik, Wilmington, DE, USA).

### 2.7. Migration Assay

Using a filter migration assay system, 5 × 10^5^ PBMCs were deposited in 200 µL (Roswell Park Memorial Institute medium) RPMI into the upper compartment containing a polycarbonate membrane filter (5 µm pore size, BD, Heidelberg, Germany). The lower compartment contained 50 ng/mL Stromal-Cell Derived Factor-1 (SDF1-α; Peprotec, Hamburg, Germany) in 500 µL RPMI or 500 µL control media. The cells were incubated at 37 °C with 5% CO_2_ for 2 h, with each sample in duplicate. The migrated cells were thereafter counted using CellTiter-Blue^®^ Cell Viability Assay (Promega). Relative migration of the cells was calculated as the difference of targeted SDF1-α migrated cells and cells migrated to control medium.

### 2.8. Statistical Analysis

Normal distribution of all data was analyzed of by Kolmogorov–Smirnov test and continuous parametric variables are presented as means ± standard derivation (mean ± SD). Luciferase assays were performed in duplicate and repeated four times. Values were compared by one-way ANOVA, followed by a Tukey post hoc test (Tukey’s multiple comparisons test) as appropriate. Gene expression analysis by real-time PCR was performed in duplicate and repeated eight times. A cDNA dilution series for AQP3 confirmed a PCR efficiency > 95%, which was comparable to the efficiency of actin. Relative AQP3 mRNA expression was measured by two-step real-time PCR with actin as the internal control and calculated as 2^−^[Ct(AQP5)−Ct(β-ctin)]. The values were compared by one-way ANOVA, two-way ANOVA, or unpaired t-test as appropriate. The characteristics of patients at baseline (timepoint of transplantation) are shown as percentages for categorical variables and as means with standard deviations (±SD) or medians with interquartile ranges (25th; 75th percentile) for continuous variables, as appropriate. Categorical variables were compared with Chi-Square or Fisher’s exact test, and continuous variables were compared with a parametric Student’s t-test or non-parametric Wilcoxon–Mann–Whitney test. Explorative comparisons based on AQP3 −1431 A/G genotypes (AG/GG vs. AA) were performed for several clinical patient characteristics (Table 1).

Acute rejection and CMV infection probabilities were graphically assessed by the Kaplan–Meier method. The log-rank test was used to evaluate the univariate relationship between the AQP3 −1431A/G genotype and incidence of acute rejection and CMV infection. In a next step, Cox regression analyses was performed, assessing the mutual effect of the AQP3 −1431A/G genotype and potential predictors on acute rejection or CMV-free survival. At first, Cox regressions were performed with several models based on a single predictor (univariate). Predictors were chosen according to current literature [18,29,30,31,32] Thereafter, multiple variable Cox regressions were performed with an initial model investigating multiple predictors simultaneously (multivariate). To avoid overfitting, a restricted model was assessed subsequently using only those predictors with a *p*-value 0.05 or lower based on either the single or multiple predictor comparisons or on Hosmer–Lemeshow test. Hazard ratio (HR) point estimates were calculated and their confidence intervals (CI) were calculated with a coverage of 95%. All reported *p*-values were nominal and two-sided with an a priori α error of <0.05. All analyses were performed using SPSS (version 25, IBM, Armonk, NY, USA); for graphical presentations, GraphPad Prism 6 (Graph-Pad Software, San Diego, CA, USA) was used.

## 3. Results

### 3.1. Characteristics of the AQP3 A(−1431)G Promoter Polymorphism

First, we carried out molecular and functional characterization of the polymorphism in vitro.

#### 3.1.1. Molecular Characteristics: AQP3 Promoter Activity and Transcription Factor Binding Based on Genotype

In a first step, we analyzed *AQP3* promoter activity and transcription factor binding in the promoter region nt-1334/nt-1491 surrounding the *AQP3* A(−1431)G promoter polymorphism.

Promoter activity in HeLa cells was nearly 40% more in promoter constructs containing A-allele compared to G-allele (*p* = 0.008; Figure 1a) and more than 50% lower in G-allele compared to empty vector (*p* = 0.034; Figure 1a). Next, we asked whether the AQP3 A(−1431)G promoter polymorphism influences transcription factor binding to the *AQP3* promoter region surrounding *AQP3* (−1431). As shown in Figure 1b, G-allele exhibited increased transcription factor binding compared to A-allele, after incubation with nuclear extracts from Caco-2 cells (similar results were obtained with extracts from other cell lines; data not shown). Hence, G-allele might facilitate binding of an inhibitory transcription factor. In silico analysis of putative transcription factor binding sites revealed SP1 and CREB on the AQP3 promoter. Thus, we investigated basal AQP3 promoter activity in altered SP1 and CREB expression.

Promoter activity of the region nt-91/nt-474 was increased by overexpressing SP1 (*p* < 0.0001) and decreased by CREB (*p* < 0.0001) (Figure 2a). Activity of the nt-1334/nt-1491 AQP3 promoter region in the G-allele promoter construct was only altered by CREB (*p* = 0.0010; Figure 2b) and not by SP1. Overall, promoter activity of A-allele was increased with SP1 and CREB expression, compared to G-allele (pcDNA3.1: *p* = 0.0483; SP1: *p*= 0.0077; CREB: *p* = 0.0272; Figure 2b).

#### 3.1.2. Functional Characteristics: AQP3 Immune Cell Expression and Migration Based on Genotype

We next examined whether the results from luciferase assays, indicating that cAMP-dependent transcription factor, CREB, reduced promoter activity in G-allele carriers, could be verified in primary human cells.

In PBMCs, AQP3 mRNA expression was four-fold higher in A-allele compared to G-allele carriers after a 2-h cAMP stimulation; however, cAMP decreased AQP3 mRNA expression in G-allele carriers more than six-fold (*p* = 0.042, Figure 3a). In addition, PBMCs from A-allele carriers exhibited target-orientated migration towards SDF1-α, while migration of PBMCs from G-allele carriers were similar towards SDF1-α and control medium (*p* = 0.037; Figure 3b). Further, AQP3 protein expression examined in different immune cells demonstrated that AQP3 protein expression was higher in T-cells compared to PBMCs and neutrophils (PBMCS vs. T-cells; *p* = 0.0057 and neutrophils vs. T-cells *p* = 0.0092, Figure 3c). T-cells from A-allele carriers showed nearly two-fold increased AQP3 expression compared to G-allele carriers (*p* = 0.012; Figure 3d).

### 3.2. Patients’ Baseline Characteristics and Course of Transplantation

To analyze potential in vivo impacts of the AQP3 polymorphism on disease course, including graft rejection and CMV infection, we analyzed clinicopathological parameters in the study patients post-transplantation. Main characteristics of patients can be found in Table 1. Frequencies for the wildtype (WT) G-allele was 63% and 37% for the variant A-allele. Regarding the distribution of genetic variations in our cohort according to the Hardy–Weinberg equilibrium of the AQP3 SNP, we observed a frequency of 85 for the GG-genotype (expected: *n* = 94), 128 for the AG-genotype (expected: *n* = 111), and 24 for the CC-genotype (expected: *n* = 32). Patients’ baseline characteristics were not associated with the AQP3 A(−1431)G polymorphism (Table 1). In addition, more than 96% of the patients received prednisone alone or in combination with ATG or Basiliximab as induction therapy, which did not differ between the AQP3 A(−1431)G genotypes (*p* = 0.672; Table 1). Furthermore, immunosuppressive maintenance therapy did not differ between genotypes (*p* = 0.681, Table 1).

#### 3.2.1. AQP3 −1431 A/G Dependent Rejection

The group of A-allele carriers contains more patients with rejection (47.3% of all A-allele carriers), compared to G-allele carriers (31.7% of all G-allele carriers) (*p* = 0.019), and the time until rejection was shorter in A-allele carriers (*p* = 0.044, Table 2).

Rejection was analyzed by Kaplan–Meyer method 30 days (early acute rejection), 180 days (late acute rejection), and 360 days after kidney transplantation. Rejection was higher in A-allele carriers compared to G-allele carriers during the entire period. As shown in Figure 4a, number of patients with acute rejection was 30% after 30 days in A-allele carriers (45/152) vs. 18% in G-allele carriers (15/85) (*p* = 0.032), 43% after 180 days in A-allele carriers (66/152) vs. 27% in G-allele carriers (23/85) (*p* = 0.009), and 45% after 360 days in A-allele carriers (69/152) vs. 27% in G-allele carriers (23/85) (*p* = 0.005).

In addition, univariate Cox analysis revealed that the AQP3(−1431) A-allele is a prognostic factor for 180-day (HR for the AA/AG genotype: 1.858, 95% CI (1.185 to 2.987), *p* = 0.011) and 360-day rejection (HR for the AA/AG genotype: 1.95, 95% CI (1.216 to 3.127), *p* = 0.006; Table 3).

This result remained robust in a multivariate analysis (360 days; Table 3) in which important risk factors reported in current literature were also considered. In addition, human leukocyte antigen (HLA)-mismatch (HR: 1.163; *p* = 0.025) and panel-reactive antibody (PRA) > 20% (HR: 2.598; *p* = 0.01) depict important risk factors for rejection; all factors remained robust in a restricted Cox-regression model (Table 4).

#### 3.2.2. AQP3 −1431 A/G Dependent CMV Infection

AQP3 A(−1431)G promoter polymorphism was not associated with CMV serology at transplantation and ganciclovir-resistant CMV strains were not detected among the study patients. One-year CMV infection risk was significantly associated with the AQP3 A(−1431)G genotypes (*p* = 0.013). CMV infection rates were 21% (32/152) for the AA/AG genotype and 35% (30/85; *p* = 0.013) for the GG genotype (Figure 4b). In addition, CMV disease was more common in individuals with the GG genotypes (11.7%; 10/85), when compared with the AA/AG genotype (4.6%; 7/152; *p* = 0.039; Table 2). Univariate Cox regression analysis revealed the genotype AQP3 (−1431) GG genotypes as an independent risk factor for CMV infection, while risk in AA/AG allele is nearly halved with a hazard ratio of 0.526 (95% CI (0.32 to 0.866); *p* = 0.012; Table 5). This result remained robust in a multivariate analysis in which known risk factors were considered and AQP3 A(−1431)G promoter polymorphism was identified as a risk factor (*p* = 0.013; Table 5), beside CMV high-risk type (D^+^/R^−^), according to CMV serology (*p* = 0.007; Table 5).

Restricted model confirmed AQP3 promotor polymorphism as the strongest and sole risk factor for CMV infection (*p* = 0.023, Table 6).

## 4. Discussion

To our knowledge, this study shows, for the first time, that the AQP3 A(−1431)G promoter SNP is associated with acute graft rejection and CMV infection after kidney transplantation. We found that this polymorphism represents a strong risk factor for acute graft rejection within one year of transplantation, and an independent and clinically meaningful risk factor of post-transplant CMV infection, with an estimated hazard ratio of nearly 2 for the corresponding genotypes. The underlying mechanisms linking altered AQP3 expression in T-cells to different AQP3 promoter activity and to immune cell migration had not been previously elucidated. In this study, the A-allele was accompanied by increased AQP3 expression in T-cells, and increased immune cell migration and conferred resistance against infection; however, A-allele carriers were prone to graft rejection. In contrast, the G-allele was associated with low incidence of graft rejection but G-allele carriers were prone to CMV infection mediated by decreased T-cell function.

Graft rejection is a common complication in kidney transplant patients [33]. As transplantation is still the most promising therapeutic option for patients with end-stage renal diseases [3], risk adapted immunomodulatory therapy is one of the key points for successful transplantation. Our study confirmed two of the most noted risk factors for rejection: HLA-mismatch and PRA [29], and further revealed AQP3 A(−1431)G promoter polymorphism as a third important risk factor in future risk prediction. Importantly, we described potential in vivo mechanisms. First, we showed that the A-allele, which is associated with increased AQP3 expression in T-cells and migration, correlated with increased graft rejection. Thus, the link between the AQP3 A(−1431)G promoter polymorphism and rejection could be due to AQP3 expression and its impact on T-cell migration. AQP3 is a well-known regulator of T-cell migration—it is suggested that AQP3-mediated water uptake is required for chemokine-dependent T-cell migration in immune response [23]. In support, a recent study demonstrated that the migration of CD4^+^ T-cells is reduced in AQP3(−/−) mice compared with that in wild type [34], and that AQP3 induced the production of some chemokines, such as chemokine (C-C motif) ligand 24 (CCL24) and CCL22 [34], which are associated with T-cell migration [35]. Furthermore, other aquaporins are well known to facilitate cell migration and impact the course of disease in several illnesses such as sepsis and cancer [17,18,36,37]. Recently, the mechanism of AQP9-induced neutrophil migration was elucidated [38] and a similar mechanism was described for AQP1 [19]. Hence, the mechanism by which AQP3 impacts migration may be similar and due to the influx of water into the cell, which creates a lower concentration of actin monomers and directs a flow of actin monomers to the site where they polymerize and form a stable lamellipodium, leading to cell migration [19]. Especially in acute graft rejection, regulatory T-cells (Tregs) play an important role by promoting a state of antigen-specific peripheral tolerance by suppressing activation and expansion of T effector cells [39,40]. In addition, memory T-cells generated against past pathogenic infections can show cross-reactivity to alloantigens, such as the HLA [41]. In our study, we could find an association of HLA-mismatch with graft rejection. Beside HLA-mismatch, our study confirmed positive pre-transplant PRA as a risk factor for acute rejection [42]. Since T-cell migration into the graft and their infiltration is essential for T-cell-mediated rejection and our study indicates that AQP3 expression is required for T-cell migration, blockade of AQP3 might depict a novel and promising therapeutic strategy to inhibit T-cell migration and could be a promising prophylactic therapy in transplantation. To date, steroids, calcineurin inhibitors, inosine-5′-monophosphate dehydrogenase inhibitors, or mTOR inhibitors are routinely applied in immune suppressive therapy [33]. However, application of these immune suppressive drugs is associated with a higher risk of infection or even cancer [43,44]. Hence, AQP3 modulation might depict a novel therapeutic option that is less prone to exerting side effects [20].

Besides graft rejection, CMV infection has an enormous effect on outcome in patients with kidney transplants and hence risk adapted, anti-CMV chemoprophylaxis is a cornerstone of modern post-transplantation management [8]. Despite high abundancy of anti-CMV prophylactic agents, CMV infections are still common [12] and consequently, better diagnostic tools—including genetic variations to stratify risk of CMV infection—and new antiviral agents with unambiguous mechanisms of action and ideally less toxicity are urgently needed. Recently we could demonstrate that the C-allele of the AQP5 −1364A/C promoter polymorphism is independently associated with an increased 12-months risk for CMV infection, which emphasize the importance of genetic variations as additional risk factors of CMV infection after solid organ transplantations [18]. Our present study revealed the AQP3 A(−1431)G promoter polymorphism as a novel promising and independent genetic risk factor for CMV infection. Based on our clinical data and considering the reduced migration of immune cells, we speculate that the altered AQP3 expression and reduced migration might shape the efficiency of immune responses, thereby influencing the efficacy of microbial clearance, and with respect to our study, CMV elimination. Due to the high complexity of the immune response to CMV infection, we can only speculate about the main AQP3 expression-related effector. However, AQP3 is highly expressed in T-lymphocytes, which play an important role in the immune response to CMV infection [45], and its expression might be essential for T-cells activity, as confirmed by the current study. In fact, T-cells showed increased AQP3 expression compared with neutrophils, and migration towards SDF1-α, a known chemokine for T-cell migration [46], depended on the AQP3 A(-1341)G SNP. Furthermore, the control of CMV infection is largely driven by adaptive immunity, involving broadly targeted CMV-specific T-cells. Patients with delayed emergence of CMV-specific CD4^+^ T-helper cells are more likely to develop a CMV infection [47]. Hence, impaired T-cell migration and reduced cytokine production, due to decreased AQP3 expression in G-allele carriers as discussed above, might be crucial for delayed T-cell function [34]. Considering these results, it can be speculated that the AQP3 A(−1431)G promoter polymorphism impacts the adaptive immune response. This hypothesis is in line with our results demonstrating that the GG genotype of the polymorphism is a strong and independent risk factor of CMV infection, as compared with the risk with AA/AG genotypes. Hence, the AQP3 A(−1431)G promoter SNP might play a pivotal role in the management of post-transplantation CMV prophylaxis.

In addition to clinical findings, our study pinpointed underlying regulatory mechanisms of the AQP3 A(−1431)G polymorphism. Substitution of A for G at position −1431 was associated with increased promoter activity of the AQP3 gene, but reduced binding of transcription factors, as shown by EMSA experiments. Thus, the G-allele may facilitate the binding of an inhibitory transcription factor. Interestingly, overexpression of the cAMP-dependent transcription factor, CREB, reduced AQP3 promoter activity in Caco-2 cells, which is in contrast to other studies where cAMP induced AQP3 expression [48]. However, this effect might be caused by inhibition of transcriptional activators, which has been previously shown for CREB and cAMP [49,50]. In addition, reduction in AQP3 expression seems to depend on the G-allele, as overexpression of CREB only reduced promoter activity of the nt-1334/-1491 promoter region in G-allele constructs and stimulation of PBMCs with cAMPs resulted in decreased AQP3 mRNA expression in G-allele carriers. Hence, the G-allele elicited decreased AQP3 promoter activity and mRNA expression compared with the A-allele. These results are attributed to decreased AQP3 protein expression in T-cells in G-allele carriers. In summary, we characterized the AQP3 A(−1341)G promoter polymorphism for the first time and showed that the G-allele binds to an inhibitory transcription factor, and elicits decreased promoter activity and decreased AQP3 RNA and protein expression in human T-cells. Furthermore, we examined the potential functional impact of the promoter polymorphism on immune cell function and showed that the G-allele was accompanied by decreased immune cell migration towards SDF1-α in vitro. Thus, our study elucidated mechanistic insights on the AQP3 SNP and transposed them to an in vivo setting by investigating graft rejection and genotype dependent CMV infection rates in kidney transplant recipients.

Nevertheless, our results might seem contradictory, as the A-allele was associated, on one hand, with lower CMV-infection rates, and on the other, with worse outcome of increased graft rejection. However, in transplantation, exaggerated immune response is associated with graft rejection; but in CMV infection, exactly the opposite must be presumed, as a more active and potent immune system can inhibit infection and have a positive effect on the risk of infection in kidney-transplanted patients.

A few limitations are evident in this investigation. First, our functional and mechanistic examinations were made in cell culture and immune cells from healthy controls, necessitating the need for in vitro experimental verification of the causality and underlying mechanisms using patient samples, which was not possible due to the lack of histologic sections. In addition, since basic promoter studies must be carried out in vitro, they depict an artificial system. However, we chose different cell lines to verify our results. In addition, EMSA experiment depicts an artificial system and the occurrence of nonspecific bands cannot be explained easily. However, we could demonstrate a specific shift, which was different between genotypes, while the concentration of the free probe remained constant. Further, our in vitro results could be confirmed by our in vivo study. In our in vivo study, a selection bias was incurred as the patients were requested to donate DNA after transplantation and specimens from patients who died were not included. Unrecognized selection bias, inherent to many genetic-association studies, cannot ultimately be excluded. Furthermore, our study was almost exclusively conducted on patients of European–Caucasian descent and, therefore, the findings cannot be generalized to subjects of other ancestries. In addition, although all patients were treated with a rather standardized multimodal regimen, we cannot exclude the possibility that unknown and potentially confounding factors exist. However, we conducted a monocentric study, where all patients were treated according to standardized protocols. Hence, differences in center specific patient populations and technical, immunosuppressive, and prophylactic regimens can be excluded. However, for the given indication, the study population was not small and multivariate Cox regression analyses revealed the AQP3 A(−1431)G polymorphism as an important and strong prognostic factor for one-year graft rejection and CMV infection.

## 5. Conclusions

In conclusion, genotype-specific modulation of AQP3 expression could be an interesting focal point for additional or rather optimized immunosuppressive and CMV prophylactic strategies after graft transplantation. However, whether this approach offers therapeutic or prophylactic benefits, needs to be elucidated in future investigations.

## Figures and Tables

**Figure 1 cells-09-01421-f001:**
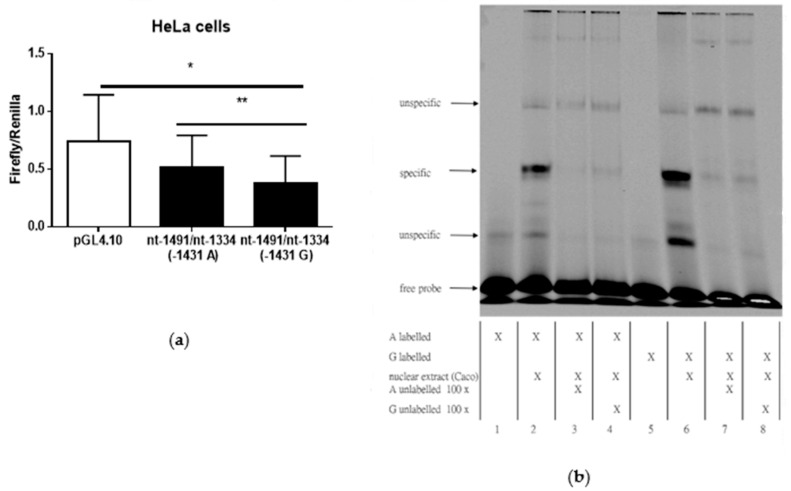
Aquaporin 3 (AQP3) promoter analysis in the promoter region of nt-1334/nt-1491 (**a**) Luciferase assay showing AQP3 promoter activity in HeLa cells (mean ± SD). In the promoter construct from nt-1334/nt-1491 activity of A-allele was compared to G-allele (*n* = 5). (**b**) Electrophoretic mobility shift assay with oligonucleotides representing the promoter region nt-1418/nt-1446 and containing the A-allele (lanes 1 to 4) and G-allele (lanes 5 to 8) of the A(−1431)G polymorphism. Representative blot of three independent experiments. The addition of nuclear extracts to labelled oligonucleotides resulted in the formation of specific bands (lanes 2 and 6). Excess of non-labelled oligonucleotide outcompeted the formation of specific bands (lanes 3, 4 and lanes 7, 8). * *p* ≤ 0.05; ** *p* ≤ 0.01.

**Figure 2 cells-09-01421-f002:**
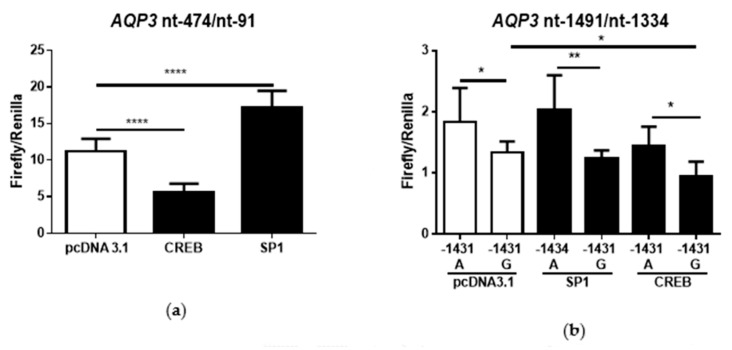
Luciferase assay showing AQP3 promoter activity (mean ± SD) in Caco-2 cells after overexpression of cAMP response element-binding protein (CREB) and specificity protein 1 (SP1). Promoter activity of the nt-91/nt-474 promoter construct (**a**) and the nt-1334/nt-1491 (**b**) was examined after co-transfection with control vector (pcDNA3.1), CREB-pcDNA3.1 (CREB), or SP1-pcDNA3.1 (SP1) vectors (*n* = 6). * *p* ≤ 0.05; ** *p* ≤ 0.01; **** *p* ≤ 0.0001.

**Figure 3 cells-09-01421-f003:**
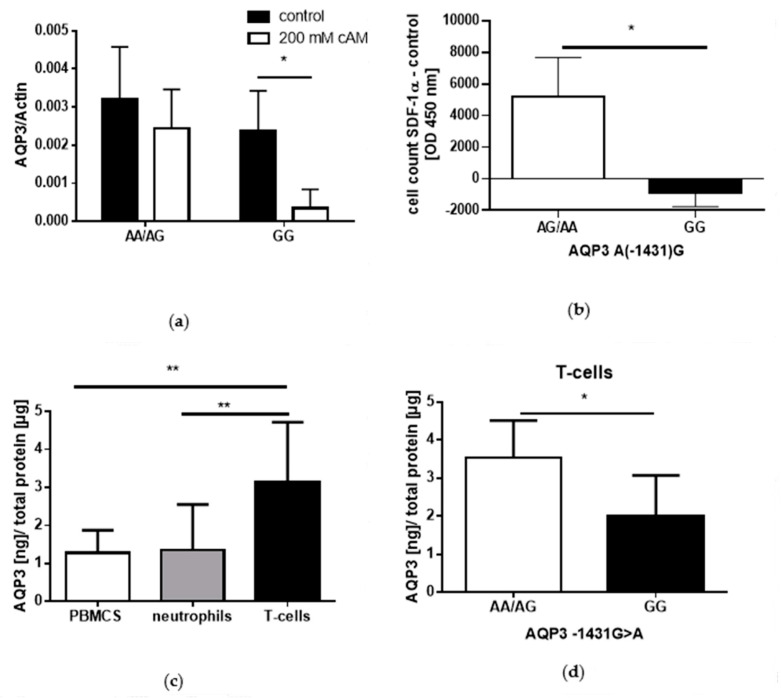
AQP3 expression and migration of immune cells depending on the AQP3 A(−1431)G genotype (mean ± SD). (**a**) shows AQP3 mRNA expression in peripheral blood mononuclear cells (PBMCs) after 8Br-cAMP (200 mM) stimulation (*n* = 6). (**b**) shows target orientated migration of PBMCs towards SDF1-α (50 ng/mL) (*n* = 8). AQP3 protein expression was examined in (**c**) PBMCs, neutrophils, and T-cells (*n* = 8) and in (**d**) T-cells depending on the genotype (*n* = 8). * *p* ≤ 0.05; ** *p* ≤ 0.01.

**Figure 4 cells-09-01421-f004:**
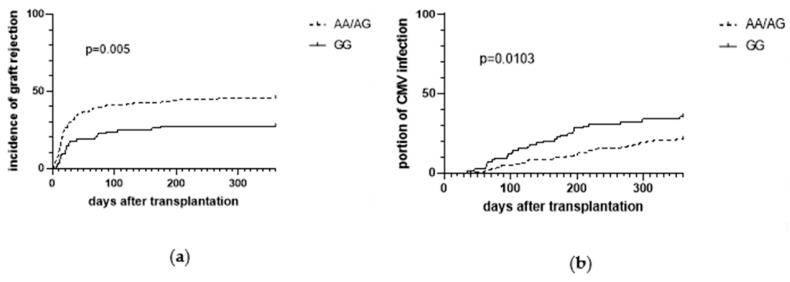
Kaplan–Meier curves showing the incidence of biopsy proven graft rejection (**a**) and the incidence of cytomegalovirus (CMV) infections (**b**) in the first year after kidney transplantation, stratified by the AA/AG and GG genotypes of the AQP3 A(−1431)G single nucleotide polymorphism (*n* = 237) as the percentual number of patients.

**Table 1 cells-09-01421-t001:** Characteristics of kidney transplant patients (*n* = 237) at baseline stratified by the AQP3 A(−1431)G genotype.

	AA/AG (*n* = 152)	GG (*n* = 85)	*p*-Value
Age (years), median (IQR)	54 (43:63)	52 (43:64)	0.257
Gender (m/w)	97/55	53/32	0.823
Height (m)	1.73 ± 0.09	1.73 ± 0.09	0.796
Weight (kg), mean ± SD	77.23 ± 14.48	77.48 ± 14.54	0.902
BMI (kg/m^2^), mean ± SD	25.48 ± 4.41	25.88 ± 4.14	0.502
**Ethnicity, *n* (%)**			0.232
Caucasian	137 (90.1%)	73 (85.9%)	
Other	15 (9.9%)	12 (14.1%)	
**Pretransplant PRA**			0.543
PRA 0–20%	144 (94.7%)	82 (96.5%)	
PRA > 20%	8 (5.3%)	3 (3.65%)	
Donor age (years), median (IQR)	52 (42.25:62)	51 (43:62.5)	0.497
Donor Gender (m/w)	79/73	47/38	0.623
Living donor, *n* (%)	15 (9.9%)	14 (16.5%)	0.137
Cadaveric donor, *n* (%)	137 (90.1%)	71 (85.5%)	
Sum of HLA-mismatches median (IQR)	3 (2:5)	3.5 (2:5)	0.522
Cold ischemia time (min), mean ± SD	705.77 ± 306.95	652.33 ± 302.26	0.201
**Immunosuppressive Induction Therapy**			0.672
Prednisone and ATG	126 (82.9%)	71 (82.7%)	
Prednisone and Basiliximab	11 (7.2%)	9 (10.6%)	
Prednisone	8 (5.3%)	4 (4.7%)	
Prednisone and ATG and Basiliximab	4 (2.6%)	0 (0.0%)	
Prednisone and Daclizumab	2 (1.3%)	1 (1.2%)	
Other	1 (0.7%)	1 (1.2%)	
**Immunosuppressive Maintenance Therapy**			0.681
Prednisone, MPA, and Calcineurin-Inhibitor	88 (57.9%)	52 (61.2%)	
Prednisone and Calcineurin-Inhibitor	23 (15.1%)	13 (15.3%)	
Prednisone, MPA and mTOR-Inhibitor	15 (9.9%)	9 (10.6%)	
Prednisone and mTOR-Inhibitor	19 (12.5%)	5 (5.9%)	
None, unknown or other	7 (4.6%)	6 (7.1%)	
**Underlying diseases**			
Diabetes mellitus	56	34	0.631
Coronary artery disease	34	15	0.389
Arterial Hypertension	107	66	0.228
Heart disease	3	5	0.110
Polycystic kidney disease	25	7	0.076
Glomerular nephritis	19	17	0.123
IgA-nephritis	9	8	0.318
Diabetic nephropathy	45	27	0.729
Interstitial nephritis	8	4	0.851
**Transplantation, *n* (%)** Kidney Combined pancreas + kidney	106 (69.7%) 46 (30.3%)	60 (70.4%) 25 (29.4%)	0.891
First kidney transplantation	136	70	0.086
Previous kidney transplantation	16	15	

IQR, Interquartile Range with 25th and 75th percentile; BMI, body mass index; HLA, human leukocyte antigen; PRA, panel reactive antibodies, ATG, antithymocyte globulin; IgA (immunoglobulin A) MPA, mycophenolic acid; mTOR, mechanistic target of rapamycin. Missing data were excluded from the analysis. Six cases were missing for body mass index and one case was missing for cold ischemia time.

**Table 2 cells-09-01421-t002:** Characteristics of kidney transplant patients (*n* = 237) at follow-up stratified by the AQP3 A(−1431)G genotype.

	AA/AG (*n* = 152)	GG (*n* = 85)	*p*-Value
Number of patients with rejection	72 (47.3%)	27 (31.7%)	*0.019*
Rejection within 1 year	69 (45.4%)	23 (27.1%)	*0.005*
**Banff Classification of Rejection within 1 Year**			0.256
Borderline	41 (59.4%)	13 (56.5%)	
Ia	14 (20.3%)	2 (8.7%)	
Ib	3 (4.3%)	1 (4.3%)	
IIa	11 (15.9%)	6 (26.1%)	
IIb	0 (0%)	1 (4.3%)	
Days until rejection, mean ± SD	68.18 ± 172.79	177.11 ± 356.92	*0.044*
CMV infection within 1 year	32 (21%)	30 (35.3%)	*0.013*
CMV disease within 1 year	7 (4.6%)	10 (11.8%)	*0.041*
**CMV Prophylaxe *n* (%)**			0.762
Valganciclovir	135 (88.8%)	74 (87.1%)	
Ganciclovir	14 (9.2%)	8 (9.4%)	
None/unknown	3 (2.0%)	3 (3.5%)	
**CMV Serology at Transplantation, *n* (%)**			0.565
D^+^/R^−^	35 (23.5%)	20 (23.5%)	
D^+/−^/R^+^	92 (61.7%)	48 (56.5%)	
D^−^/R^−^	22 (14.8%)	17 (20.0%)	
**Indication of Anti-CMV Therapy, *n* (%)**			0.909
Prophylactic–perioperative	14 (9.2%)	8 (9.4%)	
Prophylactic−3 months	100 (65.8%)	55 (64.7%)	
Prophylactic−6 months	35 (23.0%)	19 (22.4%)	
None/unknown	3 (2.0%)	3 (3.5%)	

CMV, cytomegalovirus; D^+^, CMV sero-positive donor; D^−^, CMV sero-negative donor; R^+^, CMV sero-positive recipient; R^−^, CMV sero-positive recipient; *p* ≤ 0.05 is reported in italics.

**Table 3 cells-09-01421-t003:** Multivariate Cox-regression of graft rejection over 360 days after transplantation.

Co-Variable	Univariate			Multivariate		
AQP3 −1431 Genotype	*p*-Value	HR	95% CI	*p*-Value	HR	95% CI
GG	-	1	-	-	1	-
AA/AG	*0.006*	1.95	1.216–3.127	*0.018*	1.86	1.111–3.14
Age (years)	0.857	0.99	0.983–1.015	0.522	0.99	0.97–1.016
Diabetes mellitus	0.24	1.281	0.848–1.934	0.865	1.05	0.567–1.964
BMI	0.37	1.023	0.973–1.075	0.223	1.03	0.978–1.099
**Gender**						
male	-	1	-	-	1	-
female	0.733	0.929	0.61–1.416	0.806	0.94	0.581–1.525
**Ethnicity**						
caucasian	-	1	-	-	1	-
other	0.593	1.187	0.632–2.229	0.615	1.21	0.57–2.587
HLA-mismatch	*0.025*	1.163	1.019–1.327	*0.01*	1.25	1.053–1.473
Cold ischemia time	0.706	1	0.999–1.001	0.724	1	0.999–1.001
**CMV serology**						
D^−^/R^−^	-	1	-	-	1	-
D^+^/^−^/R^+^	0.509	0.798	0.408–1.559	0.585	0.81	0.38–1.726
D^+^/R^−^	0.697	0.89	0.494–1.602	0.538	0.80	0.39–1.634
**immunosuppressive regimen**						
MPA + prednisone + tacrolimus	-	1	-	-	1	-
MPA + prednisone + cyclosporin	0.586	1.144	0.705–1.855	0.997	1.00	0.563–1.781
other	0.897	0.967	0.582–1.608	0.489	0.81	0.441–1.48
CMV infection	0.842	0.954	0.599–1.519	0.756	1.09	0.641–1.843
PRA > 20%	*0.01*	2.598	1.256–5.375	*0.002*	0.25	0.104–0.603

HR, odds ratio point estimates, 95% CI, and *p*-values (two-sided) are reported; HR, hazard ratio; HLA, human leukocyte antigen (sum of HLA-mismatches); CMV, cytomegalovirus; PRA, panel reactive antibodies. Six cases with unknown or no prophylactic anti-CMV therapy were excluded from analysis; 16 cases with missing HLA-mismatch scores were excluded from analysis. Hosmer–Lemeshow statistics for multivariable approach were as follows: κ2 = 0.47; *p* = 0.491; κ2 = 1.804; *p* = 0.179; *p* ≤ 0.05 is reported in italics.

**Table 4 cells-09-01421-t004:** Restricted multivariable Cox regression analysis of kidney transplantation recipients with respect to the effect on graft rejection risk.

Co-Variable	Multivariate Restricted		
AQP3 −1431 Genotype	*p*-Value	HR	95% CI
GG	-	1	-
AA/AG	*0.013*	1.875	1.144–3.076
HLA-mismatch	*0.006*	1.212	1.057–1.389
PRA > 20%	*0.001*	3.465	1.635–7.346

HR, hazard ratio; PRA, panel reactive antibodies, HLA, human leukocyte antigen; *p* ≤ 0.05 is reported in italics.

**Table 5 cells-09-01421-t005:** Multivariate COX-regression analysis of CMV infection over 360 days after transplantation: HR, odds ratio point estimates, 95% CI, and *p*-values (two-sided) are reported.

Co-Variable	Univariate			Multivariate		
AQP3 −1431 Genotype	*p*-Value	HR	95% CI	*p*-Value	HR	95% CI
GG	-	1	-	-	1	-
AA/AG	*0.012*	0.526	0.32–0.866	*0.013*	0.495	0.284–0.864
Age (years)	0.852	0.998	0.978–1.018	0.76	1.004	0.977–1.033
**gender**						
male	-	1	-	-	1	-
female	0.762	0.924	0.554–1.54	0.969	0.989	0.564–1.734
donor age (years)	0.69	1.003	0.988–1.018	0.992	0.999	0.978–1.02
**donor gender**						
male	-	1	-	-	1	-
female	0.139	0.686	0.416–1.131	0.303	0.751	0.435–1.295
cold ischemia time	0.872	1.000	0.999–1.001	0.654	1.000	0.999–1.001
living donor	0.277	0.603	0.242–1.503	0.323	1.788	0.565–5.652
BPAR (yes)	0.975	0.992	0.599–1.643	0.614	0.864	0.49–1.524
HLA-mismatch	0.058	1.166	0.995–1.366	0.273	1.102	0.926–1.312
**Immunosuppressive regimen**						
MPA + prednisone + tacrolimus	-	1	-	-	1	-
MPA + prednisone + cyclosporin	0.243	1.397	0.797–2.451	0.867	0.946	0.497–1.804
other	0.347	0.724	0.369–1.419	0.085	0.519	0.246–1.095
**CMV serology**						
D^−^/R^−^	-	1	-	-	1	-
D^+/−^/R^+^	0.189	2.003	0.711–5.645	0.867	0.946	0.497–1.804
D^+^/R^−^	*0.011*	3.923	1.361–11.31	*0.007*	4.541	1.515–13.612
**Anti-CMV prophylaxis**						
Ganciclovir	-	1	-	-	1	-
Valganciclovir	0.193	2.163	0.678–6.904	0.33	2.192	0.452–10.616
PRA > 20%	0.936	0.954	0.299–3.042	0.684	0.771	0.221–2.697

HR, hazard ratio; BPAR, biopsy-proven acute rejection; HLA, human leukocyte antigen; CMV, cytomegalovirus; PRA, panel reactive antibodies. Six cases with unknown or no prophylactic anti-CMV therapy were excluded from analysis; 16 cases with missing HLA mismatch scores were excluded from analysis. Hosmer–Lemeshow statistics for multivariable approach were as follows: κ2 = 1.804; *p* = 0.179; *p* ≤ 0.05 is reported in italics.

**Table 6 cells-09-01421-t006:** Restricted multivariable Cox regression analysis of kidney transplantation recipients with respect to the effect on CMV infection.

Co-Variable	Multivariate Restricted		
AQP3 −1431 Genotype	*p*-Value	HR	95% CI
GG	-	1	-
AA/AG	*0.012*	0.527	0.32–0.868
**CMV serology**			
D^−^/R^−^	-	1	-
D^+/−^/R^+^	0.152	2.134	0.756–6.022
D^+^/R^−^	*0.009*	4.089	1.418–11.795

HR, hazard ratio; CMV, cytomegalovirus; *p* ≤ 0.05 is reported in italics.

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
