# Peer review of "The Aquaporin 3 Promoter Polymorphism −1431 A/G is Associated with Acute Graft Rejection and Cytomegalovirus Infection in Kidney Recipients Due to Altered Immune Cell Migration"

_cells, 2020, doi:10.3390/cells9061421_

Round 1
Reviewer 1 Report
This was a very interesting paper to review, expanding our knowledge of the role of aquaporins in T-cell function and transplantation outcomes. I have a number of suggestions for the manuscript as detailed below.
The manuscript has clearly not been thoroughly proof-read, and my major concerns relate mainly to methodology and data presentation/analysis:
- Line 70 first introduces the rs3860987 SNP but there is no reference to the work first describing the SNP. The effects of CREB and SP1 on promotor activity are examined but no rationale is presented on why they were chosen. For the clinical study, the WT and variant allele frequencies are not reported, and Hardy Weinberg equilibrium is not documented.
- Table 1 is poorly presented and very confusing. Perhaps it would be better not to use grey and white alternating rows. Donor age and gender should be in same units as for recipient e.g. (y) and (M/F), MDRD data should be removed as it is not baseline data and is not discussed at all in the manuscript, induction with ATG (bottom of page 10) can be deleted as it is already contained in the earlier induction therapy data, CMV infection and disease should be removed as they are not baseline data, they could be presented in a separate table showing the follow-up results, CMV prophylaxis could probably also be presented as part of the follow up results and would probably be best presented as duration of prophylaxis, not just the agents used. Rejection and time to rejection should also be removed as it is not baseline data (but results) and could be presented in a separate table perhaps together with the CMV infection/disease data etc.
- I am concerned about the univariate data analyses, as this study appears to use the same patient cohort previously published by the authors examining the AQP5 SNP and CMV infection. In the earlier study, CMV risk status and duration of prophylaxis were significantly associated with CMV infection by univariate analysis, but not in this study. Could there be an error in the analysis?
- Please be consistent with the nomenclature of the promotor region, is it nt-1334/nt-1491, nt-1334 to nt-1491 or nt-1491/-1334, similarly with -474/-91. Please use the same nomenclature throughout.
- Given that the AQP5 SNP has also been shown to affect CMV infection risk in the same patient cohort, it may be interesting to look at a combined AQP3/AQP5 genotype analysis for CMV (and rejection).
Other comments:
Line 42-43: “In spite of causal therapy” change to “Despite treatment of the underlying cause”
Line 48, “high interindividual variability” in what?
Line 51, “leads to” change to “can increase the risk of”
Line 57, references 15 and 16 do not cover aquaporins, perhaps remove aquaporins (QAP) at the start of this line and change to “have been …. (15,16). Aquaporins (AQP)s affect immune cell …”.
Line 98, “Luciferase reporter assay of AQP3 promotor activity”. Rationale for CREB and SP1 should be included in this paragraph.
Line 109, “Electrophoretic mobility shift assay (ESMA) of transcription factor bidning.
Line 127, 237 patients does not agree with lines 138-142 (59+144+41=244 patients)
Figure 1 legend: a) are the data mean/median and what are the error bars, **P<??. Line 238 “unlabelled”. In the actual figure panel use A labelled and G labelled, also add Lane number (heading) at the bottom left. In panel A, it would be good to compare activity to blank vector.
Figure 2 legend, indicate significance of *, ** and ****. Be consistent with nomenclature. Again activity of blank vector should be included for comparison.
Figure 3, analysis of panel A should be 2-way ANOVA, panel C a 1-way ANOVA. Method section (line 201) indicates unpaired t-tests? How are data presented mean/media and what are the error bars?
Line 274, delete “merely”
Line 308 change “rejected organs” to “patients with acute rejection” or “rejection episodes” as appropriate.
Figure 4, Line 313 “… graft rejection (a) ..” Please check wording, it’s unclear if graphs show number of patients or number of episodes. Line 316 delete last sentence.
Line 345 (32/152) here but (33/152) in Table 1
Line 349 is not what is shown in Table 4 (consistenecy). See main comments (above) for analysis of duration of CMV prophylaxis and CMV risk.
Line 414 change “set screw” to “therapy”; Line 470 change “embank” to “inhibit”; Line 473 change “probands” to “controls”.
Discussion of CMV should contain some comparison with AQP 5 paper in the same transplant population.
Author Response
Please see the attachement.

Reviewer 2 Report
In this study, Rump and colleagues describe an association of a AQP3 promotor SNP with acute graft rejection and CMV infection risk after kidney allografting. First the authors performed a functional characterization of the AQP3 A(-1431)G SNP in vitro and subsequently tested their findings in a retrospective study of 237 patients who received kidney or pancreas-kidney transplants. After SNP analysis they performed statistical analysis of the patient's clinical data and found several risk factors associated with the AQP3 promotor variant. The risk of rejection was 45% for A allele of the AQP3 promotor, but only 27.1% for G-allele carriers. CMV incidence was 21% for A-allele carrier and 35% for G-allele carriers. The authors conclude that the A1431 allele has a lower risk for CMV infection, but seems to be associated with increased T-cell mediated graft rejection. The manuscript is well written, and the results of potential interest, I therefor recommend publication.
Author Response
Please see the attachement.

Reviewer 3 Report
- Per KDIGO report, terms "Acute or chronic kidney failure" should be avoided.
- The diagnosis of acute rejection of kidney transplant and reports need to be standardized by using The Banff Classification of Allograft Pathology in the method as well as results.
- All cases of rejection are biopsy proven? At your center, were surveillance biopsies performed?
- The diagnosis and case ascertainment of CMV needs to be provided. Organ involvement of CMV and severity of CMV disease should be provided. Protocols on CMV surveillance that the authors' institution used should be additionally provided.
- Any data on BK? does Aquaporin 3 expression affect BK occurrence?
Author Response
Please see the attachement.

Round 2
Reviewer 3 Report
I reviewed the revised manuscript and the response to reviewers' comments. Revised Manuscript is well written. All comments have been addressed and thus accepted for publication.